# Synthesis runs counter to directional folding of a nascent protein domain

Xiuqi Chen [1], Nandakumar Rajasekaran[1], Kaixian Liu[1,4] & Christian M. Kaiser [2,3✉]

Folding of individual domains in large proteins during translation helps to avoid otherwise prevalent inter-domain misfolding. How folding intermediates observed in vitro for the majority of proteins relate to co-translational folding remains unclear. Combining in vivo and single-molecule experiments, we followed the co-translational folding of the G-domain, encompassing the first 293 amino acids of elongation factor G. Surprisingly, the domain remains unfolded until it is fully synthesized, without collapsing into molten globule-like states or forming stable intermediates. Upon fully emerging from the ribosome, the G-domain transitions to its stable native structure via folding intermediates. Our results suggest a strictly sequential folding pathway initiating from the C-terminus. Folding and synthesis thus proceed in opposite directions. The folding mechanism is likely imposed by the final structure and might have evolved to ensure efficient, timely folding of a highly abundant and essential protein.

[1] CMDB Graduate Program, Johns Hopkins University, Baltimore, MD, USA. [2] Department of Biology, Johns Hopkins University, Baltimore, MD, USA. [3] Department of Biophysics, Johns Hopkins University, Baltimore, MD, USA. [4] Present address: Molecular Biology Program, Sloan Kettering Institute, New York, NY, USA. ✉email: kaiser@jhu.edu

How proteins fold into the native structures that enable their cellular functions remains a central question in biology. Folding of small proteins or single domains often follows single-exponential kinetics, suggesting a highly cooperative process[1]. However, experimental and computational studies have detected partially folded intermediates for a multitude of proteins, and proteins larger than ~100 amino acids are generally thought to populate folding intermediates before reaching their native structure[2,3].

While the presence of intermediates in most protein folding pathways is universally recognized, their functional roles are less clear. Partially structured states can represent on-pathway intermediates along a multistep folding pathway, or off-pathway misfolded states that must dissolve before productive folding can be achieved[2,4]. Distinguishing these two scenarios experimentally is often challenging, although single-molecule approaches have demonstrated their potential to provide this information[5,6].

How intermediates are linked together into a folding pathway remains a subject of debate[7,8]. The foldon hypothesis posits that small, cooperative units acquire structure in a strictly sequential order, resulting in a single folding pathway[9]. Considerations based on energy landscape theory propose that a folding protein can reach its native structure through several accessible routes[10], although specific pathways may be energetically favored. Either theory is supported by evidence from both simulations and experiments. It thus seems possible that proteins might fall into distinct categories based on their folding mechanism. Determining the relationship between amino acid sequence, folding pathway and final structure is important for designing sequences that can adopt novel structures and for understanding how natural proteins robustly reach their functional conformations.

Folding pathways are not only shaped by protein-specific properties, but also depend strongly on environmental conditions, many of which differ greatly between in vitro refolding experiments and folding in living cells[11]. The vast majority of mechanistic folding studies has been carried out with isolated proteins or domains[12]. In the cell, proteins begin to fold while they gradually emerge from the ribosome as it translates the information in the messenger RNA into a polypeptide sequence. Biophysical experiments have shown that interactions of the ribosome with the proximal part of the nascent polypeptide can reduce the stability of native domains[13,14], stabilize secondary structure inside the ribosome exit tunnel[15–17], modulate nascent chain folding kinetics[18], and prevent misfolding[18–20]. Co-translational folding is particularly important for large multi-domain proteins, because it prevents the accumulation of extended unfolded regions during synthesis that have a high propensity for misfolding[21]. Several examples show that the ribosome does not necessarily change the folding pathway[19,22,23], but the connection between vectorial emergence and folding of the nascent chain remains poorly defined.

Here, we have investigated folding of the N-terminal G-domain of *Escherichia coli* elongation factor G (EF-G). Following nascent chain folding in bacterial cells with an arrest peptide-based reporter assay[24], we find that productive folding is not initiated until the full domain has emerged from the ribosome. Force-spectroscopy experiments with optical tweezers confirmed the absence of stable structure at shorter chain lengths. These single-molecule measurements also showed that folding of the full domain proceeds through productive folding intermediates both on the ribosome and in isolation. Our studies thus reveal a strictly ordered folding pathway in which the first step requires the extreme C-terminus of the domain. Folding and synthesis of the domain therefore proceed in opposite directions.

## Results

**Detection of co-translational folding in vivo with a luciferase reporter.** To investigate folding of the G-domain from EF-G, we took advantage of the 17 amino acid arrest peptide from the *E. coli* SecM protein[25] (termed SecM17 here). While the residue at position 17 is important for arrest function, it is not incorporated into the nascent polypeptide, and SecM17 causes elongation arrest when its N-terminal 16 amino acids have been synthesized[26]. Folding-mediated release of elongation arrest results in translation of the coding sequence downstream of SecM17 and production of the full-length encoded protein[24]. The arrest peptide can thus be utilized to detect nascent chain folding that occurs in close proximity to the ribosome (Fig. 1). The assay has mainly been used to study folding in in vitro translation systems, quantifying arrest release by autoradiography[23,27–33]. To monitor folding in living cells, we utilized a genetically encoded luciferase reporter, NanoLuc[34] (Fig. 1).

To validate the in vivo folding assay, we generated reporter plasmids in which the G-domain is connected to SecM17 through 4, 17, or 30 residues of the subsequent domain II, resulting in constructs termed G+20, G+33, and G+46 (Fig. 2a). The numbers represent the sum of the domain II and SecM17 residues that separate the G-domain from the peptidyl-transferase center (PTC) at the arrest point. The G-domain (residues 1–293 of EF-G) has been shown to stably fold upon emerging from the ribosome exit tunnel[18,35], which sequesters ~30–40 residues of the nascent chain inside the large subunit of the ribosome.

The G+33 construct, which places the G-domain close to the tunnel exit, yielded a high luminescence signal upon expression in *E. coli* cells (Fig. 2b, bar diagram). At this length, the stably folded domain abuts the ribosome, generating a pulling force that destabilizes SecM arrest[24]. Consequently, translation resumes and the NanoLuc reporter is synthesized. As expected, the other two constructs yielded lower levels of reporter expression (Fig. 2b, bar diagram). In G+20, sequestration of the C-terminal ~15 amino acids in the tunnel destabilizes the G-domain; in G+46, unfolded domain II polypeptide separates the folded G-domain from the

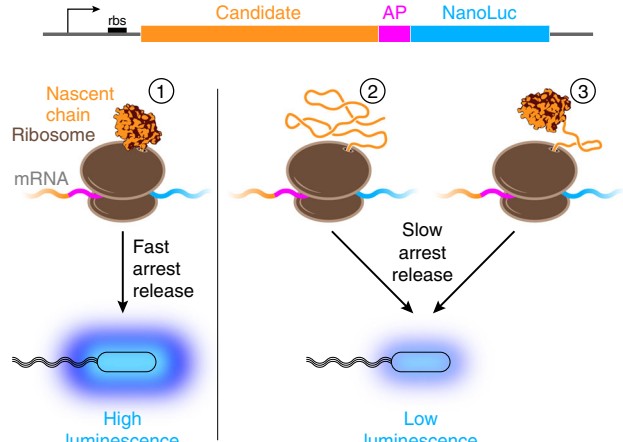

**Fig. 1 NanoLuc reporter assay for monitoring nascent chain folding in vivo.** Plasmids for expression in *E. coli* encode candidate proteins fused to the SecM arrest peptide (AP) and the NanoLuc reporter (diagram on top). Upon expression in host cells, nascent chain synthesis is arrested at the end of the AP. When the candidate is folded and in close proximity to the ribosome at the arrest point (①), force on the nascent polypeptide inside the ribosome exit tunnel accelerates arrest release. Accumulation of the full-length fusion protein that includes NanoLuc results in high luminescence. When the candidate protein is not stably folded (②) or when it is separated from the ribosome by an unfolded segment (③), arrest release occurs more slowly, and cells exhibit low luminescence.

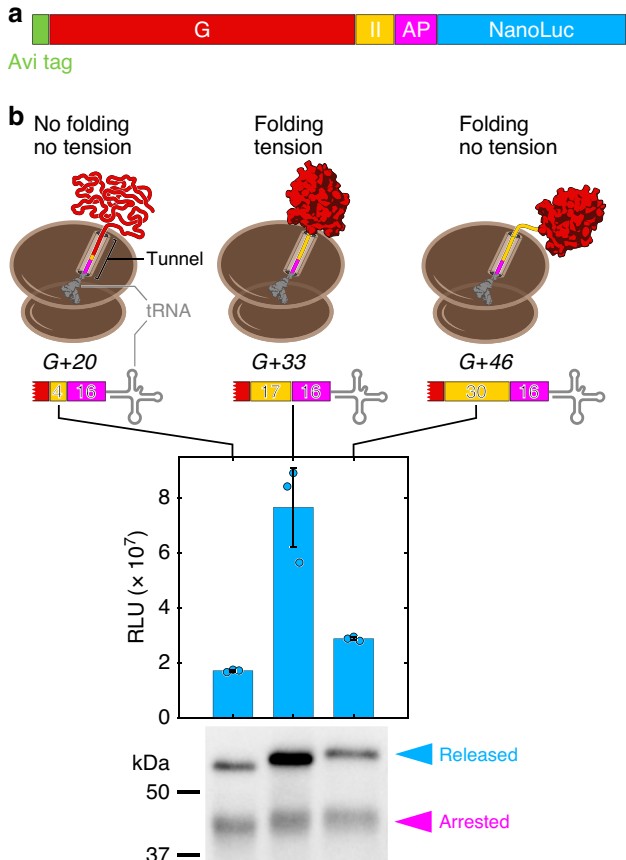

**Fig. 2 The G-domain folds upon emerging from the ribosome during EF-G synthesis. a** Construct design for measuring folding of the G-domain (G) in live cells. The G-domain is extended by a variable number of domain II (II) residues in the folding reporter construct (see also Fig. 1). **b** Arrest release measurements for three distinct lengths of EF-G nascent chains. Luminescence measurements (bars: mean; circles: individual data points) show that arrest release is high in the G+33 construct, compared to shorter (G+20) or longer (G+46) nascent chains. Cartoons (top) illustrate the positioning of the G-domain (red), the domain II segment (yellow), and the arrest peptide (magenta) relative to the tRNA in the ribosome P-site. Western blot analysis (bottom panel) shows similar amounts of arrested nascent chains (magenta arrowhead), but clear differences in the accumulation of the full-length fusion protein (blue arrowhead), confirming that the luminescence signal is due to accelerated arrest release. Error bars represent the standard deviation from three independent experiments. RLU relative light units.

ribosome, preventing the generation of tension on the nascent chain (Fig. 2b, cartoons).

To verify that the luminescence readouts in our experiments reflect increased NanoLuc accumulation, rather than reduced specific activity of NanoLuc caused by fusion to poorly folded polypeptides[36], we visualized arrested and full-length translation products by Western blotting. We observed similar levels of arrested protein for all three constructs (magenta arrowhead in Fig. 2b). After arrest release, ribosomes continue to elongate and synthesize the luciferase reporter, resulting in accumulation of full-length protein over time. The significantly higher amount of full-length product for G+33 compared to the control constructs (blue arrowhead in Fig. 2b) therefore reflects an increased arrest release rate. This result indicates that elevated luminescence indeed reports on folding-mediated release of SecM arrest.

**Colony luminescence identifies arrest-releasing candidates.** The folding of some small proteins or domains is well described by a two-state model, while proteins of the size of the G-domain usually populate intermediates during folding[2,3]. To determine whether intermediates are formed co-translationally, we generated 75 individual reporter constructs with EF-G inserts ranging in length from 72 to 368 amino acids (Fig. 3a). When the ribosome stalls at the SecM sequence, the separation of the N-terminal G-domain residue from the PTC of the ribosome, referred to here as length $L$ (Fig. 3b), ranges from 88 to 384 aa (16 residues from SecM plus 72 and 368 EF-G residues, respectively).

We transformed *E. coli* cells with a mixture of all plasmids at similar concentrations and grew colonies on inducing agar. We applied the luciferase substrate to the plate and imaged the resulting colony luminescence (Fig. 3c). Analyzing the light emission of individual colonies allowed us to distinguish highly luminescent colonies from nonluminescent (dark) colonies (Supplementary Fig. 1). Sequencing revealed that constructs yielding highly luminescent colonies all encoded candidate proteins with lengths ranging from 308 to 328 amino acids (Fig. 3d, cyan dots). It thus appears that colony luminescence reliably identifies arrest-releasing constructs in the region where full folding of the G-domain is expected. Nonluminescent colonies, selected from a small area of the plate that also contained highly luminescent colonies (Fig. 3c, blowup), all had candidate lengths outside this region (Fig. 3d, gray dots). This result suggests that, surprisingly, no stable intermediates are formed co-translationally, and that only full G-domain folding constitutes a major folding waypoint during EF-G synthesis.

**Full in vivo folding profile of the G-domain.** The on-plate assay provides a convenient format for analyzing a pool of candidates in a simple experiment. However, colony luminescence only provides a binary readout, and subtle differences in signal are not resolved. To compare arrest release rates more quantitatively, we assayed the 75 EF-G truncation constructs individually by growing them separately in liquid cultures. After normalizing by cell density, culture luminescence intensity shows a clear peak with the maximum at $L = 332$ (Fig. 4). Visualization of the arrested and released product in the region of $280 \leq L \leq 340$ by Western blotting (Supplementary Fig. 2) confirmed that luciferase activity reports on arrest release. Strong arrest release is therefore detected when the C-terminal G-domain residue (aa 293) is separated from the PTC by 39 aa, a value similar to that observed for other relatively large domains[31].

Notably, a signal increase to ~25% of the maximum intensity coincides with completion of G-domain synthesis ($L = 309$). At this length, offset from the maximum of the peak by 23 amino acids (Fig. 4, vertical bar), all of the G-domain except for the C-terminal helix has been extruded from the ribosome. The slightly elevated arrest release rates in this length range could suggest the formation of meta-stable structures outside the ribosome[29,32], or formation of secondary structure inside the exit tunnel[16,31,37]. All chain lengths shorter than 308 amino acids exhibit only basal luciferase activity (below 20% of the peak value; gray box in Fig. 4), suggesting that the G-domain does not form stably folded intermediates. Given that the G-domain is large, this finding is unexpected.

**Stably structured states are not formed until synthesis is complete.** Several folding scenarios can explain why most of the G-domain truncations do not exhibit luminescence above the baseline level. If partially folded structures are destabilized by the ribosome, they may become stably folded only after an intervening spacer has been synthesized, preventing detection of these

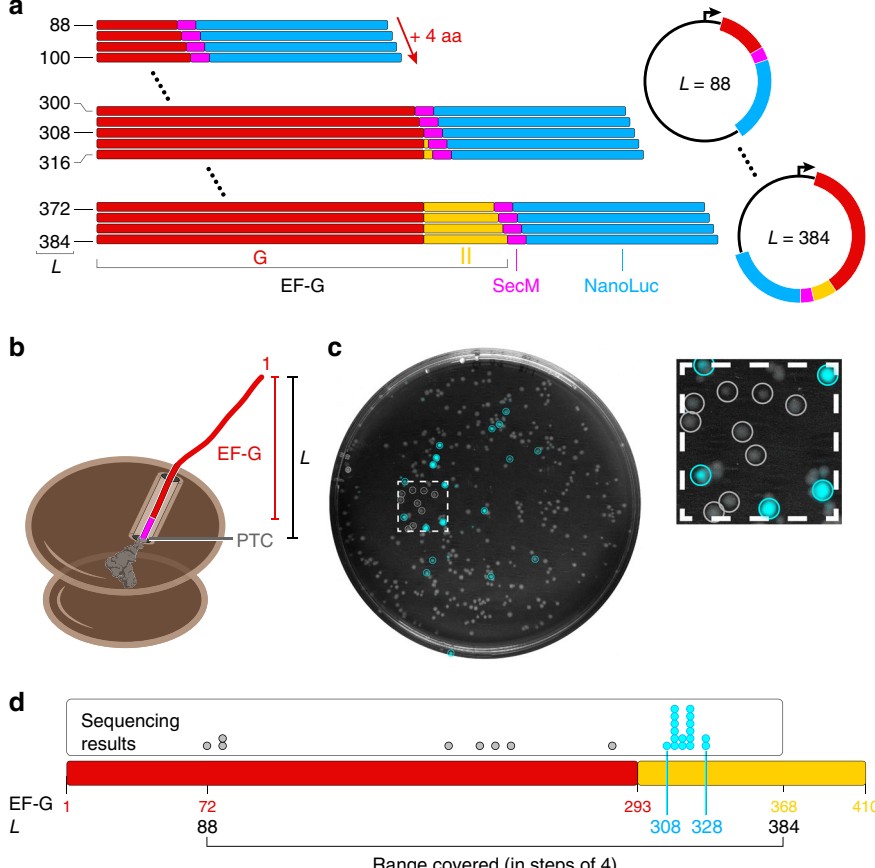

**Fig. 3 On-plate screening identifies nascent chain folding in a pool of constructs. a** Construction of a plasmid pool containing increasingly longer EF-G candidate polypeptides. The shortest construct encodes 72 N-terminal amino acids of the G-domain (red), followed by the SecM arrest sequence ($L = 88$); the longest construct encodes the full G-domain plus 75 amino acids from domain II (yellow), followed by the SecM arrest sequence ($L = 384$). **b** Cartoon illustrating the nomenclature for the plasmid pool. The length $L$ represents the number of amino acids from the peptidyl-transferase center (PTC) of the ribosome to residue 1 of the EF-G candidate. **c** Merged brightfield image of the agar plate (grayscale) and colony luminescence (cyan). The inset shows a magnification of the boxed region. Colonies selected for sequencing are circled. **d** Colony sequencing results. Each dot represents the sequencing result from an individual colony, mapped to the length of the EF-G candidate sequence (bar diagram). All constructs yielding high luminescence (cyan dots) cluster in the region $308 \leq L \leq 328$. All nonluminescent samples have shorter chain lengths (gray dots). Red and yellow numbers indicate residue positions in the EF-G amino acid sequence.

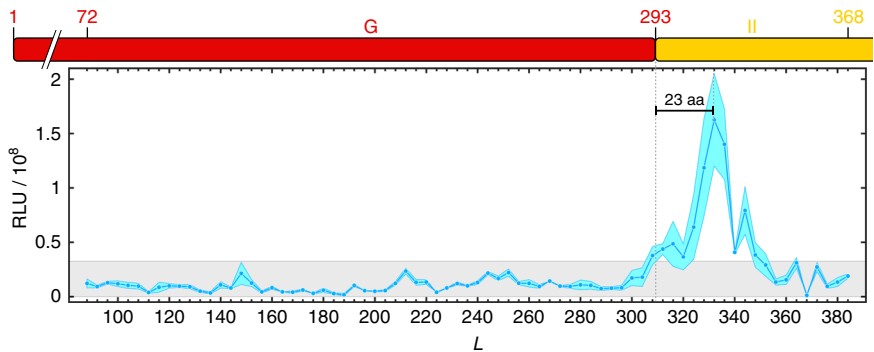

**Fig. 4 Folding profile of EF-G nascent chains.** The plot shows the luminescence readings from 75 individual cultures. The blue line indicates the mean from three independent measurements, with the standard deviation indicated by the cyan-colored area. A strong peak is centered around $L = 332$, where the G-domain has been extruded from the ribosome. Shorter and longer constructs outside the main peak region show only background luminescence (gray box indicates 20% of maximum signal), suggesting the absence of folding intermediates before synthesis of the entire domain is complete. The EF-G domain diagram on top is drawn for reference to indicate synthesis progress. Sequence positions (domain diagram, top) are offset from $L$ (x-axis of the plot) by 16 amino acids, the length of the SecM arrest peptide that is incorporated into the nascent polypeptide.

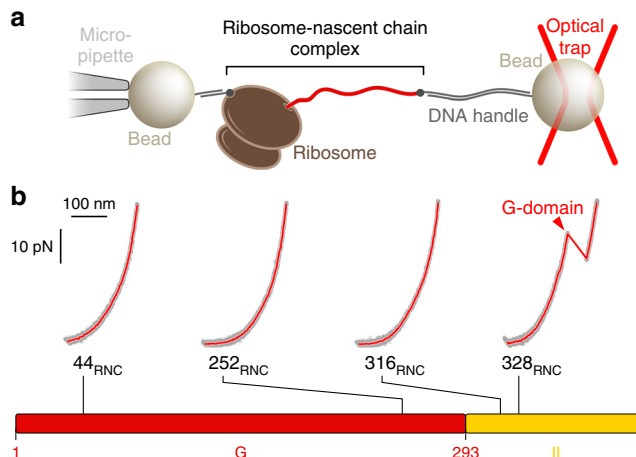

**Fig. 5 Nascent chain structure probed with single-molecule force spectroscopy. a** Cartoon schematic (not to scale) of the experimental setup. A single ribosome-nascent chain complex (RNC) is tethered between two polystyrene beads by means of DNA handles. Moving the bead held in the optical trap enables application of force to nascent EF-G polypeptides. **b** Representative force-extension curves of RNCs with four distinct lengths (gray dots: raw data, 1000 Hz; red lines: filtered data, 30 Hz). When the G-domain is fully outside the ribosome ($328_{RNC}$), the characteristic unfolding transition of the G-domain is observed (red arrowhead). Shorter chain lengths ($44_{RNC}$, $252_{RNC}$, $316_{RNC}$) do not show defined unfolding transitions. The bar diagram at the bottom indicates the positions of the chain lengths in the EF-G primary structure.

intermediates in the in vivo reporter assay. Alternatively, the G-domain may only begin to form stable structures once the complete domain has been synthesized and is mostly exposed to the outside of the ribosome.

To distinguish between these scenarios, we probed the structure of nascent EF-G polypeptides directly using single-molecule force spectroscopy with optical tweezers (Fig. 5). We generated terminally stalled ribosome-nascent chain complexes (RNCs) by in vitro translation of nonstop messenger RNAs[18]. Tag sequences at the N-terminus of the nascent chain and on protein L17 in the large ribosomal subunit allowed us to tether these complexes for mechanical manipulation with optical tweezers (Fig. 5a). Mechanical force acts as a denaturant that destabilizes folded structures. In force ramp experiments, a continuously increasing tension is applied to the tethered molecule. In the resulting force-extension curves, unfolding of nascent chain structure results in rips, sudden increases in molecular extension as the polypeptide transitions from a compact folded to an extended unfolded state. The method is thus suitable to detect tertiary structure in individual protein molecules.

In RNCs with a length of $L = 328$ (termed $328_{RNC}$ here), unfolding of the native G-domain results in a characteristic transition (Fig. 5b), as observed previously[18,35]. As expected, a very short nascent chain ($44_{RNC}$) that exposes only a few EF-G residues outside the exit tunnel does not exhibit any transitions in these experiments. Surprisingly, however, substantially longer nascent chains ($252_{RNC}$ and $316_{RNC}$) do not exhibit defined unfolding transitions, either (Fig. 5b). Occasionally, we detect heterogeneous transitions at these chain lengths (Supplementary Fig. 3). Their distribution is distinct from that expected for unfolding of well-defined states and might be reminiscent of misfolded states that have been observed with other proteins[19,38]. Regardless of what these heterogeneous transitions represent, the results from our single-molecule experiments indicate that the

nascent G-domain does not form stable folding intermediates or collapsed states, even when almost all of its sequence has emerged from the ribosome.

**Multistate folding of the full domain follows a strict order.** To follow folding of the ribosome-bound G-domain, we carried out optical tweezers experiments with $328_{RNC}$ in force clamp mode. In these experiments, the force is held at a constant value while changes in molecular extension are recorded. After fully unfolding the G-domain, we jumped the force to 3.5 pN to initiate refolding. The molecule transitions repeatedly between the unfolded and partially folded states (Fig. 6a), reflecting the complexity in folding that is expected for a large domain. Transitions cease upon forming a fully structured stable state (Fig. 6a, open arrowhead). The partially structured states exhibit several distinct extensions that may represent productive on-pathway folding intermediates or misfolded off-pathway species. Folding to the native state occurs from a partially folded structure (Fig. 6a, gray arrowhead), demonstrating the presence of at least one on-pathway intermediate.

A complex pattern of folding intermediates is also observed in force clamp experiments with the isolated G-domain in the absence of the ribosome (Fig. 6b). Previous work[18] showed that the overall folding rate of $328_{RNC}$ is reduced compared to that of the isolated G-domain, implying interactions of the nascent chain with the ribosome prior to folding. The ribosome may therefore affect the stability of folding intermediates or transitions between them. Nevertheless, the intermediates exhibit similar extensions in the isolated G-domain and the ribosome-bound nascent chain, suggesting similar folding pathways in both scenarios. As observed for $328_{RNC}$, the final folding step to the stably structured domain initiates from a partially structured state (Fig. 6b, gray arrowhead).

The on-pathway folding intermediate is ~10 nm more extended than the natively folded G-domain (Fig. 6a, b, gray vs. open arrowheads). At the refolding force of 3.5 pN, this length corresponds to ~90 amino acids of unfolded polypeptide, as calculated using a worm-like chain model[39] with a persistence length of 0.65 nm and a contour length of 0.36 nm per amino acid. The intermediate would therefore be composed of ~200 structured G-domain residues. Transitions consistent with unfolding of such a structure are observed in some of the force ramp recordings obtained with $328_{RNC}$ (Supplementary Fig. 3, dashed gray line). The absence of similar transitions in shorter nascent chains ($252_{RNC}$ and $316_{RNC}$, Fig. 5b and Supplementary Fig. 3) suggests that amino acids near the C-terminus of the G-domain are required for the formation of the obligatory on-pathway intermediate. Consistent with this observation, constant force measurements with the $316_{RNC}$ nascent chain do not exhibit compaction into a partially structured state (Fig. 6c and Supplementary Fig. 4). It thus appears that the G-domain remains largely unfolded during synthesis until its C-terminus, encompassing the last alpha-helix of the domain, has been extruded from the exit tunnel. Taken together, our measurements are consistent with a strictly ordered folding pathway that begins at the extreme C-terminus of the G-domain.

## Discussion

We have defined the folding pathway of the nascent G-domain from EF-G using a combination of in vivo (Figs. 2–4) and single-molecule measurements (Figs. 5 and 6). Notably, no stable folding intermediates are detected in vivo or in vitro until the complete domain of 293 amino acids is synthesized. Previously, the small src SH3 domain (64 amino acids) was found to fold only upon reaching the exit of the ribosome tunnel[22]. However, polypeptides

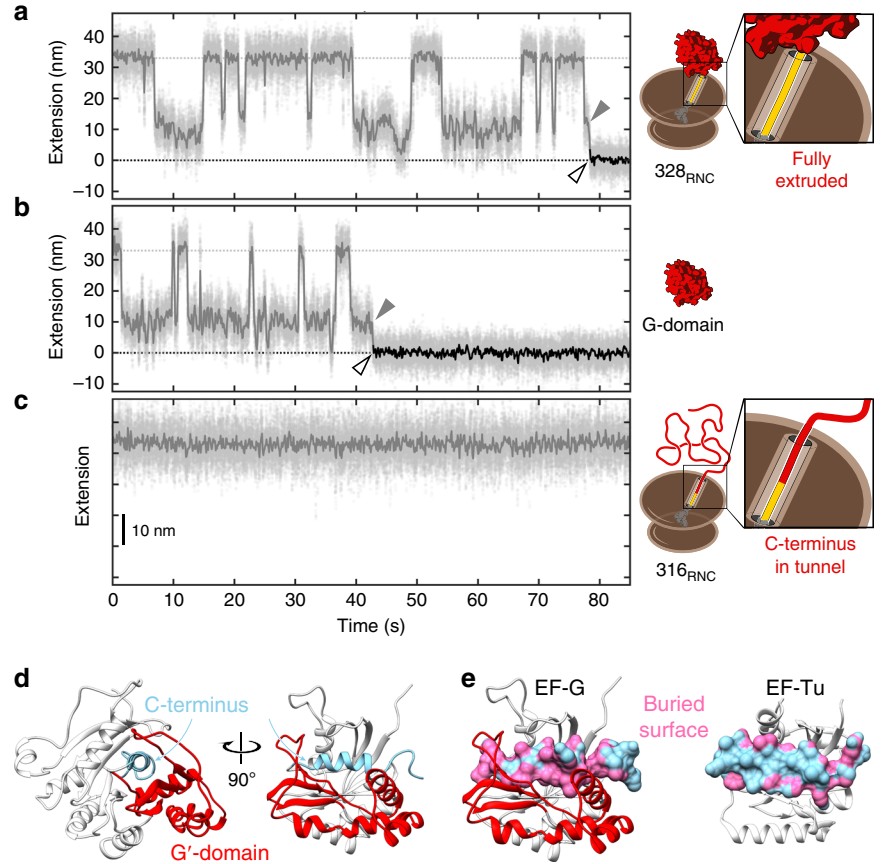

**Fig. 6 Refolding of the G-domain on and off the ribosome. a** Refolding of the ribosome-bound G-domain (328$_{RNC}$). Starting with the unfolded G-domain at $t = 0$, refolding against a constant force of 3.5 pN is followed by recording changes in molecular extension. The domain visits several partially folded states with extensions between the unfolded (gray dotted line) and folded (black dotted line) states. At least one state represents an on-pathway folding intermediate (gray arrowhead) that is visited immediately before the domain reaches the folded state (open arrowhead). Gray dots: raw data (1000 Hz), solid lines: filtered data (30 Hz). **b** Same as **a** for the isolated G-domain. The overall trajectory resembles that of the ribosome-bound G-domain. **c** Same as **a** for a shorter nascent chain (316$_{RNC}$). In contrast to 328$_{RNC}$, the shorter nascent chain does not exhibit clear folding transitions, suggesting that the C-terminal amino acids of the G-domain, which remain confined within the exit tunnel at this chain length, are essential for formation of productive folding intermediates. **d** Cartoon representation of the G-domain. The G′ insertion (red) and the remaining part of the G-domain (white) enclose the C-terminal region (light blue) containing the last α-helix. **e** G-domains from EF-G (left) and EF-Tu (right) in the same orientation as the right panel in **d** with the C-terminal region shown in surface representation. The surface area of this region that is buried in the native structure is shown in pink. Whereas the C-terminus is mostly buried in EF-G, the face closest to the viewer is largely accessible in EF-Tu. Structural representations were generated with Chimera[55] using pdb coordinates 4v9p (EF-G) and 6eze (EF-Tu).

composed of more than 100 amino acids are commonly assumed to populate folding intermediates[2,3]. It is thus surprising that the N-terminal ~250 amino acids of the G-domain do not appear to acquire stable structures.

Consistent with previous results[18,35], the G-domain forms a stable structure after fully emerging from the ribosome ($L = 328$), which manifests as a large peak in the SecM reporter assay (Fig. 4). Nonrandom signal fluctuations at shorter chain lengths ($48 < L < 320$) might represent secondary structure formations, especially α-helices, which have previously been shown to accelerate SecM arrest release[29,31,40]. The identity of nascent chain sequences inside the ribosome exit tunnel has been described to also affect arrest release[29,31,32]. Interactions of nascent chain residues with the ribosome could prevent formation of the SecM secondary structure that is required to cause stalling and could thus in principle account for the observed signal. Regardless of their origin, the amplitudes of these fluctuations are small compared to the main peak, suggesting the absence of stably folded structures until the G-domain is fully synthesized.

Single-molecule optical tweezers experiments confirm that folding begins only after synthesis is complete. We do not detect well-defined stable structures in nascent chains at lengths up to $L = 316$ (Fig. 5). Small, heterogeneous transitions that are occasionally observed for the incomplete G-domain (Supplementary Fig. 3) suggest misfolded states that are likely suppressed in vivo by molecular chaperones. Optical tweezers measurements readily detect partially folded (e.g., in alpha-synuclein[38] and T4 lysozyme[19]) or collapsed, molten globule-like structures (e.g., in ribonuclease H[41] and apomyoglobin[42]). The contact order[43] of the G-domain is moderately low (relative contact order: 0.07), and its overall hydrophobicity[44] is similar to that of other proteins (Supplementary Fig. 5). These factors therefore do not appear to account for the observed lack of early structure formation or collapse during synthesis. Its sequence properties might help to keep the G-domain extended and prevent (premature) collapse into kinetically trapped states[45]. Ribosome interactions, previously shown to destabilize native[13,46] and non-native[19,35] nascent chain structures, might further contribute to

keeping the nascent domain unfolded until its synthesis is complete.

Interestingly, our measurements suggest a relatively sharp transition in the propensity to form compact structures as the nascent chain is elongated. At $L = 316$, the nascent chain has properties of an intrinsically disordered protein, whereas the addition of just 12 amino acids to $L = 328$ results in the formation of collapsed or partially structured states and subsequent folding to the native structure (Figs. 5 and 6). Theoretical studies concluded that the formation of compact states is an evolved property of natural proteins[47]. The G-domain may be an attractive model to investigate how this property is related to protein sequence and structure.

Our studies provide an example of folding occurring in the direction opposite to that of synthesis and contrast with previous findings of gradual compaction and folding concomitant with protein elongation[48]. Decoupling of folding and synthesis has previously been observed. Folding of the N-terminal regions of the low-density lipoprotein receptor (LDL-R) in the endoplasmic reticulum is delayed by the formation of intermediates that are stabilized by non-native disulfide bonds, which slowly rearrange into the native configuration[49]. Thus, the protein completes its folding post-translationally. LDL-R folds in the oxidizing environment of the endoplasmic reticulum, whereas the G-domain emerges from the ribosome into the cytosol, remaining unfolded. Once the full G-domain has emerged from the ribosome, folding occurs in several steps that appear similar on the ribosome and in isolation (Fig. 6). The ribosome therefore does not seem to change the folding pathway. The late onset of folding and the detection of defined intermediates suggest that folding proceeds along a sequential pathway. This scenario is consistent with the foldon hypothesis, in which well-defined states are formed in a prescribed order[50].

The strict sequentiality of G-domain folding might be dictated by the final structure. The region containing the C-terminal helix is largely buried in the folded structure (Fig. 6d, e). Perhaps folding must occur with the helix serving as a central nucleus around which the remainder of the structure is formed subsequently, rather than by inserting the helix into preformed intermediates. Interestingly, part of this enclosure around the C-terminal helix is formed by the G′ domain, an insertion present in some, but not all G-domain containing elongation factors[51] (Supplementary Fig. 6). In future studies with homologous G-domains (such as those shown in Supplementary Fig. 6), it will be interesting to examine whether lack of the G′ insertion allows intermediate formation during synthesis and relaxes the strict order of folding that we observe here for EF-G. Nascent G-domains appear as attractive models for investigating how folding pathways co-evolve with structures that enable crucial cellular functions.

EF-G is a highly abundant protein (top 1% in the *E. coli* proteome[52]) that fulfills an essential cellular function. The efficiency of EF-G synthesis and folding may have been under evolutionary pressure. The coding sequence contains very few rare codons (Supplementary Fig. 7), suggesting that it is translated without major pauses[53]. Collapsed states and intermediates can kinetically trap folding proteins in non-native states[4]. Structure acquisition through a strictly ordered sequential pathway upon completion of synthesis might have evolved as a mechanism to ensure timely folding of EF-G.

## Methods

**Bacterial strains, plasmids, and reagents**. In vivo folding experiments were carried out in *E. coli* strain Lemo21(DE3) (New England Biolabs, NEB, C2528J). All plasmids used in this study are based on a backbone with a pUC origin, Lac-operator-controlled T7 promoter, and Ampicillin resistance gene[35]. The NanoLuc coding sequence was obtained as a synthetic DNA fragment (Integrated DNA Technologies). The SecM coding sequence was introduced through synthetic DNA fragments that served as primers for PCR amplification. Vector backbone and PCR products were assembled with Gibson Assembly Master Mix (NEB E2611), yielding plasmid pWP3. All primers used for cloning are listed with their sequences in Supplementary Information (Tables 1–3). The amino acid sequence of the arrest peptide used in this study is FSTPVWISQAQGIRAGP. All commercially available enzymes were purchased from NEB unless stated otherwise. PCR reactions were carried out with Phusion high-fidelity DNA polymerase (Thermo Scientific, F530S). Chemicals were purchased from Sigma-Aldrich unless stated otherwise. Streptavidin-HRP for Western blot detection was from SouthernBiotech™ (#7100–05).

**Construction of EF-G truncation library**. EF-G fragments of defined lengths were individually PCR-amplified with a NheI tailed universal forward primer (WP3-EF-G-uni-fwd) and reverse primers at designated positions along the EF-G open reading frame (WP3-EF-G-44 to WP3-EF-G-424; see Supplementary Table 2 for all primer sequences). The backbone containing the SecM-NanoLuc sequence was amplified from pWP3 with a SpeI tailed forward primer (WP3-bb-fw) at the AviTag and reverse primer at the SecM (WP3-bb-rv). After PCR clean-up, backbone and insert were mixed at a 1 to 3 ratio (40 fmol in total) in CutSmart® Buffer (NEB) supplemented with 3-mM ATP and 10-mM DTT. The reaction was provided with 4U of each NheI, SpeI, T4PNK and T4 DNA ligase at 10 μl final volume. After 2 h at 37 and 16 °C overnight, the product was transformed and verified by colony PCR with Taq DNA polymerase. Plasmid DNA from colonies showing the correct insert sizes was isolated, and its correct sequences were verified by Sanger sequencing.

**On-plate NanoLuc assay**. Cells were transformed with indicated plasmids per manufacturer instructions and spread on a LB agar supplemented with ampicillin, chloramphenicol, 500 μM L-rhamnose and 500 μM IPTG on a 9 cm diameter round plate. Colonies were grown at 37 °C for 12–16 h. We reconstituted 500 μl Nano-Glo® Live Cell Reagent (Promega N2011) for each plate and sprayed evenly onto the plate with an airbrush (Neo Iwata HP-CN N4500). Images were taken in a shaded home-made imaging box equipped with a Canon Rebel T3 camera, operated in raw image acquisition mode to avoid complications from camera-internal image processing. Camera settings were at Neutral between 5 and 15 s exposure and ISO800 for optimal contrast. Images were recorded with 8-bit color depth. To identify nascent chains that fold into stable structures, colony luminescence was quantified using custom Matlab scripts for image analysis. Circular areas of identical size (shown in Fig. 3c) were defined around colonies of interest, and integrated intensities were obtained by summing the intensities values of all pixels within these circles (see Supplementary Fig. 1). Colonies with an integrated intensity above 20,000 were designated as luminescent, colonies with integrated intensities below 10,000 were designated as nonluminescent (dark). For sequence analysis, we picked colonies that were well separated on the plate to avoid cross contamination between colonies. Dark colonies chosen for analysis were from an area of the plate that also contained highly luminescent colonies to rule out that uneven coating of the plate with luciferase substrate accounted to the lack of luminescence. DNA was amplified from selected colonies (circles in Fig. 3c) by PCR, and the resulting PCR products were analyzed by Sanger sequencing.

**Liquid culture NanoLuc assay**. Cells transformed with individual plasmids were spread on LB agar with antibiotics and allowed to form colonies overnight. LB supplemented with ampicillin and chloramphenicol was inoculated with a single colony to grow an overnight culture. Overnight cultures were diluted into fresh LB containing antibiotics to $OD_{600} = 0.01$ and incubated in a 37 °C shaker at 220 rpm. Cell densities were monitored with a plate reader (ThermoMax Microplate Reader, Molecular Devices). At $OD_{600} = 0.2–0.4$, cultures were induced with 500 μM L-rhamnose and 500 μM IPTG for 1 h. Cell densities were measured and 100 μl cultures were put onto a white round-bottom 96-well plate for luminescence measurement. All NanoLuc assays in this study were carried out with Nano-Glo® Live Cell Assay from Promega (N2011) according to manufacturer instructions. Luminescence was measured on a GloMax® Navigator Microplate Luminometer (Promega). Signals were linearly normalized to $OD_{600} = 0.4$. Each data set was acquired using identical instrument settings to allow comparison between samples. The integration time chosen such that the highest signal did not exceed 1E9 RLU to avoid saturating the detector. For visualization of translation products by Western blotting, Streptavidin-HRP was used to detect biotinylated proteins in whole-cell lysates after ribonuclease treatment. Uncropped images of the Western blots shown in Fig. 2 and Supplementary Fig. 2 are provided in the Source Data file.

**Single-molecule force spectroscopy with RNCs**. Stalled RNCs were generated as described previously[18,35]. Nonstop mRNA templates were generated by in vitro transcription of PCR products (see Supplementary Table 3 for primer sequences). Stalled RNCs were produced by in vitro translation, isolated by ultracentrifugation, and dissolved in HKMβ buffer (20 mM HEPES, 100 mM KCl, 5 mM MgCl₂, 5 mM β-mercaptoethanol, pH 7.4), and stored in small aliquots after flash-freezing. Single-molecule experiments were carried out using a custom home-built

instrument[54]. The experiments were carried out in HKMβ buffer. For force ramp experiments, the trap was moved at a constant speed of 100 nm/s to apply continuously increasing forces on the nascent chain in the range from 2 to 50 pN. The force was ramped down at the same loading rate, and the molecule was held at 2 pN for 10 s before being pulled again. Data were collected at a sampling frequency of 1000 Hz and averaged to 33 Hz for plotting. The extension changes were determined using custom MATLAB scripts as described in detail previously[18]. For constant force experiments, the molecule was first subjected to force ramp cycles to ensure it exhibited the characteristic unfolding transitions. After fully unfolding the molecule at 30 pN, the force was reduced to 3.5 pN to initiate refolding. The change in molecular extension was recorded at a sampling frequency of 1000 Hz. The measurement does not yield the absolute extension of the molecule. For measurements with $328_{RNC}$ and with the isolated G-domain, the extension of the folded state was defined as 0, all other extensions are relative to this reference state. For measurements with the unfolded $316_{RNC}$, the relative extension is reported.

**Reporting summary**. Further information on research design is available in the Nature Research Reporting Summary linked to this article.

## Data availability
PDB Accession Codes for data sets used in Fig. 6 and Supplementary Fig. 6 are 4V9P (EF-G), 6EZE (EF-Tu), 2H5E (RF3), and 3CB4 (LepA). All data are available from the corresponding author upon reasonable request. Source data are provided with this paper.

## Code availability
The custom Matlab code that was used for data analysis in this study is available from the corresponding author upon reasonable request.

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

## Acknowledgements

C.M.K. acknowledges support from the National Institutes of Health (5R01GM121567) and from the Pew Biomedical Scholars Program.

## Author contributions

C.M.K. and X.C. designed research, X.C. and N.R. generated reagents, X.C., N.R., and K.L. performed experiments, all authors analyzed data, C.M.K., X.C., and N.R. wrote the manuscript, all authors edited the manuscript.

## Competing interests

The authors declare no competing interests.
