## [Peer Review File · Nature Communications]

REVIEWER COMMENTS

Reviewer #1 (Remarks to the Author):

In this manuscript, Chen et al. reported an interesting finding that the G-domain of elongation factor G folds only after its C-terminal region exists from the ribosome exit channel. Using a fluorescence-based co-translational folding assay, they first demonstrated that the C-terminal region is required for G-domain folding in vitro. Then they confirmed this conclusion in vivo using various truncated G-domains at their C-terminus. Finally, the authors pulled single G-domain alone or attached to a ribosome using optical tweezers. Although the G-domain folds via an intermediate, they found that folding of the intermediate requires the C-terminal region. Overall, the manuscript is well-written, data are beautiful, and the major conclusion is convincing and interesting. To my knowledge, the work demonstrates a new paradigm of co-translational protein folding. The following comments should be addressed during revision of the manuscript:

1. In the western blot shown at the bottom of Fig. 2, it is not clear why the arrested proteins have similar intensities, but the released proteins greatly differ. Did ribosome continue to synthesize proteins once the arrested proteins are rescued by G-domain folding?
2. In Fig. 3A, the yellow region (linker to SecM) is only shown at the bottom diagram, but missing in all top diagrams. Should the yellow region be included in all top diagrams as well?
3. In Fig. 6, it would be good to draw a diagram for the intermediate folding state that includes a highlighted C-terminal helix (corresponding to the light blue region in C-D?).

Reviewer #2 (Remarks to the Author):

This manuscript by Chen et al. reports on co-translational folding behaviors of elongation factor G (EF-G) from *E. coli*. The authors used the in vivo force sensing assay (Figure 1 to 4) and the single-molecule optical tweezers (Figure 5 and 6) to convincingly show that there are no stable intermediates along the folding pathway of EF-G and that the final native structure is markedly stabilized by the addition of the very C-terminal amino acid (a.a.) sequence. Importantly however, these results do not necessarily mean that the folding process occurs from the C- to the N-terminus, which is currently poised as the main conclusion of the manuscript. The folding pathway can take many arbitrary routes to form less stable intermediates, and such alternative scenarios equally explains the data as long as the addition of the very C-terminal sequence (estimated to be as short as 12 a.a. residue according to the data in Figure 5) completes the native structure. This yet unresolved issue, directly related to the main conclusion, precludes recommendation of the current manuscript for publication in *Nature Communications*. As discussed below, however, the single-molecule experiment the authors have designed has the potential for directly addressing this issue.

Major comments:

1. The results from the two assays indicate that the structural stability of EF-G is markedly increased with the addition of the very C-terminal amino acid sequence, which is in line with many previous reports including one from the Bustamante group (*Nature* (2010) 465, 637-640). In the folding traces shown in Figure 6, the initial largely-wandering traces stably reside in a single state over tens of seconds once they arrive in a compact state assigned as the native folded state. As discussed above, these observations do not necessarily mean that the folding does not start at all until the C-terminal sequence is translated and added. In fact, there are minor peaks observed in the in vivo force-sensing assay (Figure 4, the minor peaks discernible at $L \approx 150$, ~ 210 and ~ 250). More

importantly, the authors observed a clear on-pathway intermediate that is approximately 10 nm displaced from the native state at 3.5 pN (Fig. 6A,B), which strongly suggests formation of a major structure that involves more than 200 a.a. residues. This intermediate is only marginally stabilized, thus short-lived and lasting only a few seconds under 3.5 pN tension.

The current manuscript concludes that this intermediate is formed by the 200 a.a. residues from the C-terminal end and that the final folding step corresponds to addition of the remaining N-terminal residues. However, it seems equally possible that the intermediate is formed by the N-terminal 200 residues with addition of the C-terminal sequence dramatically consolidating the structure. The marginal stability of the intermediate may account for the failures in observing a substantial release of the translation arrest (Figure 3 and 4) and detection of any clear unfolding transitions in the force-ramp experiments (Figure 5).

The issue might be directly addressed by conducting the force-clamp experiments in Figure 6 for the 252RNC and 316RNC constructs used in Figure 5. It would be important to see whether these slightly shorter constructs exhibit the same on-pathway intermediate observed in Figure 6. If no intermediates are observed, that would strongly support the authors' conclusion that the folding is indeed initiated from the C-terminal end. However, if the same intermediates are observed for these shorter constructs (while still failing to reach the native state), the data would indicate that the more N-terminal parts work to form this on-pathway intermediate, which is later stabilized by addition of the very C-terminal sequence.

2. A modified version of the *in vivo* force-sensing assay was used for the experiments in Figure 2 to 4, in which a luciferase reporter was fused downstream to the strong arrest sequence of SecM (17 residue long) such that release of the arrest led to translation of the luciferase reporter. A basic control experiment was carried out in Figure 2, confirming the previous reports that the largest pulling force was generated when EF-G was displaced from the peptidyl transferase center by ~30 amino acid residues. While this experiment in Figure 2 used the full-length sequence of EF-G with varying lengths of the linker region, the experiments in Figure 3 and 4 used the truncated EF-G sequence in probing the degree of arrest release. Since the truncated EF-G proteins would form marginally stable intermediates at best and thus remain largely unfolded for most of time, it is important to check in an additional control experiment whether the luciferase reporter equally works in a designed way even when fused to truncated polypeptides.

Minor comments

1. The SecM arrest sequence used in this work is 17 residue long and should thus be derived from *E. coli*. The authors need to confirm this information and specify this in the manuscript.
2. Figure 1 simply shows the scheme for the experiment in Figure 2. The authors may want to consider merging Figure 1 and 2 into one.
3. In the method part, more experimental details need to be specified for how the authors separated the high- and the low-luminescence colonies in the experiment of Figure 3.
4. The construct design in Figure 3D shows a pink region at the N-terminal end, which is not shown in Figure 3A.

Reviewer #4 (Remarks to the Author):

In this manuscript, Xiuqi Chen and co-authors attempted to determine the pathway of the cotranslational folding of the elongation factor G (namely, its "small" G-domain, encompassing the first 293 amino acids). They have used a combination of *in vivo* and single molecule experiments to look for folding intermediates. The authors followed folding of nascent protein chains of different lengths (short; extended just to allow G-domain extrusion out of the ribosome tunnel; and extended far beyond the tunnel exit) expressed in bacterial cells. They have utilized SecM arrest peptide-based reporter assay (the folding of a nascent chain just outside the tunnel is expected to generate a pulling force sufficient to unblock the arrest) to determine the folding pathway *in vivo* and found that productive folding is not initiated until the full domain has emerged from the ribosome. Subsequent

force spectroscopy experiments with optical tweezers showed the absence of stable structure at shorter chain lengths. Their result suggested that no stable intermediates are formed cotranslationally and that the initiation of the folding of the G-domain requires the extreme C-terminus of the domain. It appeared that the domain remains unfolded until it is fully synthesized, without collapsing into molten globule-like states or forming stable intermediates. The authors concluded that folding and synthesis of the domain proceed in opposite directions.

This is a well-executed study adding to our understanding of the mechanism(s) of the protein folding in the cell.

1. The authors however overlooked a similar study by Braakman and colleagues, entitled "Coordinated Nonvectorial Folding in a Newly Synthesized Multidomain Protein, which also showed that the most amino-terminal domain (of the protein under investigation, the low-density lipoprotein receptor (LDL-R)) acquired its native conformation late in folding instead of during synthesis (Science 2002). The authors need to discuss and compare their results with the results of the above mentioned paper.

2. The N-terminal ~250 amino acids of the G-domain do not appear to acquire any stable structures. The authors speculated that structure acquisition through a strictly ordered sequential pathway upon completion of synthesis might have evolved as a mechanism to ensure timely folding of EF-G. They have suggested that such pathway is facilitated by ultra fast translation (as the coding sequence contains very few rare codons). The authors have to use ribosome profiling to prove the point and look at the dynamics of ribosome movement.

3. Trigger Factor (TF) is known to influence the force profile by generally reducing the force exerted on the new protein and slowing the final folding conformation. The lead author of the paper previously used fluorescence spectroscopy to observe in real-time the actions of TF on translating ribosomes and found that it interacts with both the ribosome and the nascent polypeptide to prevent protein misfolding and aggregation. I wonder what is the role of TF in the folding of G-domain? The authors used Lemo21(DE3) strain which contains TF. What would be the outcome of these experiments in the strain lacking TF?

Response to reviewers' comments

We are grateful for the constructive feedback from the reviewers on our manuscript "Synthesis runs counter to directional folding of a nascent protein domain". In our point-by-point response below, reviewer comments are shown in black, our response is shown in blue font color. In quoted text from the manuscript, changes and insertions are highlighted in red.

Reviewer #1:

In this manuscript, Chen et al. reported an interesting finding that the G-domain of elongation factor G folds only after its C-terminal region exists from the ribosome exit channel. Using a fluorescence-based co-translational folding assay, they first demonstrated that the C-terminal region is required for G-domain folding in vitro. Then they confirmed this conclusion in vivo using various truncated G-domains at their C-terminus. Finally, the authors pulled single G-domain alone or attached to a ribosome using optical tweezers. Although the G-domain folds via an intermediate, they found that folding of the intermediate requires the C-terminal region. Overall, the manuscript is well-written, data are beautiful, and the major conclusion is convincing and interesting. To my knowledge, the work demonstrates a new paradigm of co-translational protein folding. The following comments should be addressed during revision of the manuscript:

1. In the western blot shown at the bottom of Fig. 2, it is not clear why the arrested proteins have similar intensities, but the released proteins greatly differ. Did ribosome continue to synthesize proteins once the arrested proteins are rescued by G-domain folding?

The reviewer raises an important point. The bands representing SecM-stalled nascent chains in Fig. 2B indeed show similar intensities. This observation suggests that the steady state amount of arrested ribosomes is similar for all three constructs, and that release of SecM arrest is the rate-limiting step for synthesizing the reporter-containing protein. After a SecM-arrested ribosome has overcome arrest, the ribosome continues to elongate and synthesizes the luciferase reporter. The stall site is then occupied by another ribosome that was cued upstream of the arrest site on the mRNA. As such, the band intensity likely reflects the amount of mRNA present. The similar band intensities would thus suggest that the mRNA concentrations are similar for all three constructs. Because the full-length product (which includes the luciferase reporter) accumulates over the course of the experiment, its intensity in the Western blot is much higher in G+33 than in the two other constructs, reflecting the higher luminescence reading. To better point out this important aspect, we have made the following change to the main text:

"We observed similar levels of arrested protein for all three constructs (magenta arrowhead in Figure 2B). After arrest release, ribosomes continue to elongate and synthesize the luciferase

reporter, resulting in accumulation of full-length protein (blue arrowhead in Figure 2B) over time. The significantly higher amount of full-length product for $G+33$ compared to the control constructs (blue arrowhead in Figure 2B) therefore reflects an increased arrest release rate. This result indicates that elevated luminescence indeed reports on folding-mediated release of SecM arrest.” (p. 3)

2. In Fig. 3A, the yellow region (linker to SecM) is only shown at the bottom diagram, but missing in all top diagrams. Should the yellow region be included in all top diagrams as well?

Prompted by the reviewer’s comment, we revisited Fig. 3A and found it to display the intended concept in a confusing manner. The panel meant to convey the lengths of the EF-G sequence insertions upstream of the SecM arrest peptide, which range from 72 to 368 amino acids (aa) in steps of 4 aa. The yellow region represents part of domain II (aa 294 to 410 in EF-G), which begins to be part of the construct at a length of $L = 312$ aa). We have revised the figure to better reflect this aspect and hope that it better explains the experimental design.

3. In Fig. 6, it would be good to draw a diagram for the intermediate folding state that includes a highlighted C-terminal helix (corresponding to the light blue region in C-D?).

We appreciate the reviewer’s suggestion and agree that showing a diagram of the folding intermediate would enhance Figure 6. While our data indicates that the C-terminal alpha-helix is an essential part of the on-pathway folding intermediate, we do not have information about the exact structure of the intermediate. Given that folding proceeds from the intermediate directly to the folded state of the G-domain, it is likely that the intermediate contains a large fraction of native structure. However, depicting its structure would be highly speculative. We therefore respectfully refrain from drawing a diagram of the intermediate structure. However, we have modified the cartoon in Figure 6 (panels A and C) to better point out that our findings suggest the importance of the C-terminus for G-domain folding, which is fully exposed in 328RNC (panel A) and partially buried in 316RNC (panel C).

Reviewer #2:

This manuscript by Chen et al. reports on co-translational folding behaviors of elongation factor G (EF-G) from *E. coli*. The authors used the in vivo force sensing assay (Figure 1 to 4) and the single-molecule optical tweezers (Figure 5 and 6) to convincingly show that there are no stable intermediates along the folding pathway of EF-G and that the final native structure is markedly stabilized by the addition of the very C-terminal amino acid (a.a.) sequence. Importantly however, these results do not

necessarily mean that the folding process occurs from the C- to the N-terminus, which is currently poised as the main conclusion of the manuscript. The folding pathway can take many arbitrary routes to form less stable intermediates, and such alternative scenarios equally explains the data as long as the addition of the very C-terminal sequence (estimated to be as short as 12 a.a. residue according to the data in Figure 5) completes the native structure. This yet unresolved issue, directly related to the main conclusion, precludes recommendation of the current manuscript for publication in Nature Communications. As discussed below, however, the single-molecule experiment the authors have designed has the potential for directly addressing this issue.

Major comments:

1. The results from the two assays indicate that the structural stability of EF-G is markedly increased with the addition of the very C-terminal amino acid sequence, which is in line with many previous reports including one from the Bustamante group (Nature (2010) 465, 637-640). In the folding traces shown in Figure 6, the initial largely-wandering traces stably reside in a single state over tens of seconds once they arrive in a compact state assigned as the native folded state.

As discussed above, these observations do not necessarily mean that the folding does not start at all until the C-terminal sequence is translated and added. In fact, there are minor peaks observed in the in vivo force-sensing assay (Figure 4, the minor peaks discernible at $L \sim 150$, ~ 210 and ~ 250). More importantly, the authors observed a clear on-pathway intermediate that is approximately 10 nm displaced from the native state at 3.5 pN (Fig. 6A,B), which strongly suggests formation of a major structure that involves more than 200 a.a. residues. This intermediate is only marginally stabilized, thus short-lived and lasting only a few seconds under 3.5 pN tension.

The current manuscript concludes that this intermediate is formed by the 200 a.a. residues from the C-terminal end and that the final folding step corresponds to addition of the remaining N-terminal residues. However, it seems equally possible that the intermediate is formed by the N-terminal 200 residues with addition of the C-terminal sequence dramatically consolidating the structure. The marginal stability of the intermediate may account for the failures in observing a substantial release of the translation arrest (Figure 3 and 4) and detection of any clear unfolding transitions in the force-ramp experiments (Figure 5).

The issue might be directly addressed by conducting the force-clamp experiments in Figure 6 for the 252RNC and 316RNC constructs used in Figure 5. It would be important to see whether these slightly shorter constructs exhibit the same on-pathway intermediate observed in Figure 6. If no intermediates are observed, that would strongly support the authors' conclusion that the folding is indeed initiated from the C-terminal end. However, if the same intermediates are observed for these shorter constructs (while still failing to reach the native state), the data would indicate that the more N-terminal parts work to form this on-pathway intermediate, which is later stabilized by addition of the very C-terminal sequence.

The reviewer raises a very important point. We interpreted the minor peaks as arising from either the formation of fleeting secondary structure formation or interactions of

nascent chain residues (upstream of the SecM sequence) with the tunnel (see Discussion, p. 6). However, we agree with the reviewer that the force ramp measurements shown in Figure 5 do not conclusively rule out that incompletely extruded nascent chains form the obligatory on-pathway intermediate observed with 328RNC and the isolated G-domain, or another folding intermediate of similar stability. We therefore followed the reviewer's suggestion and carried out force clamp experiments. Because the current pandemic situation makes it challenging to carry out experiments in the lab, we focused on the 316RNC construct for these experiments. We reasoned that any intermediates that might be formed by the shorter 252RNC construct would also be present in the longer 316RNC construct.

The new data has been incorporated as panel C in Figure 6 and Supplementary Figure 4. The recordings do not show any significantly stable structures formed during the length of individual recordings (approximately 100 seconds for most traces). This result supports the original conclusion that the C-terminus is required for intermediate formation. We hope that the new experimental data directly addresses the valid concern raised by the reviewer. The new result is described in the Results section:

“The absence of similar transitions in shorter nascent chains (252_{RNC} and 316_{RNC}, Figure 5B, and Supplementary Figure 3) suggests that amino acids near the C-terminus of the G-domain are required for the formation of the obligatory on-pathway intermediate. Consistent with this observation, constant force measurements with the 316_{RNC} nascent chain do not exhibit compaction into a partially structured state (Figure 6C, and Supplementary Figure 4). It thus appears that the G-domain remains largely unfolded during synthesis until its C-terminus, encompassing the last alpha-helix of the domain, has been extruded from the exit tunnel.” (p. 5-6)

2. A modified version of the in vivo force-sensing assay was used for the experiments in Figure 2 to 4, in which a luciferase reporter was fused downstream to the strong arrest sequence of SecM (17 residue long) such that release of the arrest led to translation of the luciferase reporter. A basic control experiment was carried out in Figure 2, confirming the previous reports that the largest pulling force was generated when EF-G was displaced from the peptidyl transferase center by ~30 amino acid residues. While this experiment in Figure 2 used the full-length sequence of EF-G with varying lengths of the linker region, the experiments in Figure 3 and 4 used the truncated EF-G sequence in probing the degree of arrest release. Since the truncated EF-G proteins would form marginally stable intermediates at best and thus remain largely unfolded for most of time, it is important to check in an additional control experiment whether the luciferase reporter equally works in a designed way even when fused to truncated polypeptides.

The reviewer is correct that fusion of the luciferase reporter to long unstructured or unstable proteins has the potential to interfere with reporter activity, e.g. by causing misfolding of the reporter, causing it to be enzymatically inactive. This phenomenon was

observed and exploited by Waldo et al. (reference 36 in the original manuscript), where misfolding of a fusion partner resulted in reduced GFP fluorescence. To rule out that the incomplete G-domain similarly interferes with NanoLuc activity, we carried out the suggested control experiment. We repeated the expression of reporter constructs ranging from L = 280 to L = 340 and analyzed the resulting samples by Western blotting. This data, now included as Supplementary Figure 2, shows a good correlation between the reporter activity and accumulation of released full-length protein. The new data thus shows that low luciferase before the main folding peak is indeed due to stable SecM arrest, rather than accumulation of inactive reporter, even in the presence of long stretches of unfolded polypeptide. The reporter therefore appears to faithfully report on folding-mediated release of SecM arrest.

The new data is described in the Results section:

“Visualization of the arrested and released product in the region of $280 \leq L \leq 340$ by Western blotting (Supplementary Figure 2) confirmed that luciferase activity reports on arrest release.” (p. 4)

Minor comments

1. The SecM arrest sequence used in this work is 17 residue long and should thus be derivated from *E. coli*. The authors need to confirm this information and specify this in the manuscript.

We apologize for the incomplete description. We have revised the manuscript to point out in the Results section that we are using the *E. coli* SecM sequence and are now providing the amino acid sequence in the Methods section:

“... we took advantage of the 17 amino acid arrest peptide from the *E. coli* SecM protein²⁵ (termed *SecM17* here).” (p. 2)

“The amino acid sequence of the SecM17 arrest peptide used in this study is “FSTPVWISQAQGIRAGP”.” (p. 8)

2. Figure 1 simply shows the scheme for the experiment in Figure 2. The authors may want to consider merging Figure 1 and 2 into one.

We thank the reviewer for the suggestion. It is correct that Figure 1 shows the scheme of the arrest release experiment. It does apply not only to Figure 2, but also to Figures 3 and 4. We considered merging Figures 1 and 2, but think that the original arrangement as separate figures better serves the purpose of explaining the general concept of the assay.

3. In the method part, more experimental details need to be specified for how the authors separated the high- and the low-luminescence colonies in the experiment of Figure 3.

We thank the reviewer for pointing out this shortcoming. We have created a new figure (Supplementary Figure 1 in the revised manuscript) that shows the separation of high- and low-luminescence colonies and added more detail to the Methods section:

“To identify nascent chains that fold into stable structures, colony luminescence was quantified using custom Matlab scripts for image analysis. Circular areas of identical size (shown in Figure 3C) were defined around colonies of interest, and integrated intensities were obtained by summing the intensities values of all pixels within these circles (see Supplementary Figure 1). Colonies with an integrated intensity above 20,000 were designated as luminescent, colonies with integrated intensities below 10,000 were designated as “dark”. For sequence analysis, we picked colonies that were well separated on the plate to avoid cross-contamination between colonies. Dark colonies chosen for analysis were from an area of the plate that also contained highly luminescent colonies to rule out that uneven coating of the plate with luciferase substrate accounted to the lack of luminescence. DNA was amplified from selected colonies (circles in Figure 3C) by PCR, and the resulting PCR products were analyzed by Sanger sequencing.” (p. 8 – 9)

4. The construct design in Figure 3D shows a pink region at the N-terminal end, which is not shown in Figure 3A.

We apologize for the confusion. We intended to display the region of EF-G not covered in the screen (amino acids 1 to 71 of EF-G) as semi-transparent. However, the region looks pink in the rendered figure. We have removed the transparent region in Figure 3D for clarity.

Reviewer #4:

In this manuscript, Xiuqi Chen and co-authors attempted to determine the pathway of the cotranslational folding of the elongation factor G (namely, it's “small” G-domain, encompassing the first 293 amino acids). They have used a combination of in vivo and single molecule experiments to look for folding intermediates. The authors followed folding of nascent protein chains of different lengths (short; extended just to allow G-domain extrusion out of the ribosome tunnel; and extended far beyond the tunnel exit) expressed in bacterial cells. They have utilized SecM arrest peptide-based reporter assay (the folding of a nascent chain just outside the tunnel is expected to generate a pulling force sufficient to unblock the arrest) to determine the folding pathway in vivo and found that productive folding is not initiated until the full domain has emerged from

the ribosome. Subsequent force spectroscopy experiments with optical tweezers showed the absence of stable structure at shorter chain lengths. Their result suggested that no stable intermediates are formed cotranslationally and that the initiation of the folding of the G-domain requires the extreme C-terminus of the domain. It appeared that the domain remains unfolded until it is fully synthesized, without collapsing into molten globule-like states or forming stable intermediates. The authors concluded that folding and synthesis of the domain proceed in opposite directions. This is a well-executed study adding to our understanding of the mechanism(s) of the protein folding in the cell.

1. The authors however overlooked a similar study by Braakman and colleagues, entitled “Coordinated Nonvectorial Folding in a Newly Synthesized Multidomain Protein, which also showed that the most amino-terminal domain (of the protein under investigation, the low-density lipoprotein receptor (LDL-R)) acquired its native conformation late in folding instead of during synthesis (Science 2002). The authors need to discuss and compare their results with the results of the above mentioned paper.

We apologize for the oversight and thank the reviewer for pointing out this reference, which is included in the revised version of the manuscript (reference 49). It is indeed interesting to compare the results from the Braakman group (for LDL-R) with those of our study (for the EF-G G-domain). For both LDL-R and the G-domain, folding is delayed relative to translation. In the case of LDL-R, formation of intermediates with non-native disulfide bonds is a prominent feature of the folding pathway, whereas the G-domain appears to remain largely unstructured in our experiments until it has fully emerged from the ribosome. We now discuss this aspect in the manuscript:

“Decoupling of folding and synthesis has previously been observed. Folding of the N-terminal regions of the low-density lipoprotein receptor (LDL-R) in the endoplasmic reticulum is delayed by the formation of intermediates that are stabilized by non-native disulfide bonds, which slowly rearrange into the native configuration⁴⁹. Thus, the protein completes its folding post-translationally. LDL-R folds in the oxidizing environment of the endoplasmic reticulum, whereas the G-domain emerges from the ribosome into the cytosol, remaining unfolded.” (p. 7)

2. The N-terminal ~250 amino acids of the G-domain do not appear to acquire any stable structures. The authors speculated that structure acquisition through a strictly ordered sequential pathway upon completion of synthesis might have evolved as a mechanism to ensure timely folding of EF-G. They have suggested that such pathway is facilitated by ultra fast translation (as the coding sequence contains very few rare codons). The authors have to use ribosome profiling to prove the point and look at the dynamics of ribosome movement.

In the original version of the manuscript, we suggested that translation of the EF-G coding sequence is “swift” (p. 7 in the original manuscript), based on the absence of rare codon clusters (Supplementary Figure 7 in the revised manuscript). We did not

mean to imply that translation is ultra-fast. We have modified the relevant sentence to better point out that fact:

“The coding sequence contains very few rare codons (Supplementary Figure 7), suggesting **that it is translated without major pauses**⁵³.” (p. 7)

We also appreciate the suggestion to analyze ribosome profiling data for detecting the presence or absence of pause sites as a proxy of translation speed along the EF-G coding sequence. While we do not have the resources to carry out ribosome profiling experiments ourselves, we used a recently published dataset from the Green and Buskirk labs (Mohammad *et al.*, *Cell Reports* 2016). Figure R1, appended to this document, shows the densities of ribosome footprints along the EF-G coding sequence, along with the corresponding A-site positions. The open reading frame of EF-G does indeed show relatively low ribosome occupancy, in line with the suggested absence of pauses. The highest peaks are seen near the 5' end of the open reading frame. For comparison, the neighboring rpsG open reading frame shows more pronounced peaks, which could indicate translational pausing at defined positions. Taken together, our preliminary analysis of available ribosome profiling data suggests that major translational pause sites are absent from the EF-G coding sequence.

3. Trigger Factor (TF) is known to influence the force profile by generally reducing the force exerted on the new protein and slowing the final folding conformation. The lead author of the paper previously used fluorescence spectroscopy to observe in real-time the actions of TF on translating ribosomes and found that it interacts with both the ribosome and the nascent polypeptide to prevent protein misfolding and aggregation. I wonder what is the role of TF in the folding of G-domain? The authors used Lemo21(DE3) strain which contains TF. What would be the outcome of these experiments in the strain lacking TF?

The reviewer raises intriguing questions. TF has indeed been shown to reduce the force on nascent chains and also to delay their folding. Since TF is super-stoichiometric to ribosomes, it is reasonable to assume that it associates with most translating ribosomes and, as such, also interacts with the nascent G-domain. Interactions with TF do not appear required to keep the G-domain in an unfolded state during synthesis, as our single-molecule experiments carried out in the absence of the chaperone indicate that incomplete chains do not form stable structures. These experiments also indicate that TF is not strictly required for folding of the complete G-domain, at least under the dilute conditions of our *in vitro* experiments. Nevertheless, TF could help to avoid aggregation and/or misfolding in the crowded cellular environment. We agree with the reviewer that experiments with a TF deletion strain would be very informative. Following up on a previous study of TF function in EF-G folding (Liu *et al.*, *Mol Cell* 2019), we are currently preparing to carry out *in-cell* and single-molecule experiments to further define the function of TF in EF-G folding, which will be the subject of a future study.

Figure R1. Screenshot from GWIPS-viz browser showing ribosome profiling results for the EF-G coding sequence. The region shown here includes the *fusA* gene, encoding EF-G, and a neighboring gene, *rpsG*, encoding a ribosomal protein. The approximate boundaries of the G-domain in *fusA* are indicated by a red box. The bottom track shows the footprints of elongating ribosomes, the upper track shows their A-site positions. The highest density within the *fusA* region is observed near the 5' end of the coding sequence. Downstream regions do not show prominent peaks that would indicate pronounced pausing. For comparison, the adjacent *rpsG* coding sequence exhibits stronger peaks that may indicate pausing during translation. Data from Mohammad *et al.*, *Cell Rep* 14(4): 686, 2016.

REVIEWERS' COMMENTS

Reviewer #1 (Remarks to the Author):

The authors have carefully revised the manu and addressed my concerns.

Yongli Zhang

Reviewer #2 (Remarks to the Author):

The authors addressed the reviewers' comments in a thorough manner with additional experimental data, which is remarkable given the current pandemic situation. In particular, the new observations that the construct 316RNC does not show a stable folding intermediate even under a prolonged observation time of ~ 300 s (Figure 6C and Suppl. Figure 4) shore up the authors' conclusion that the folding process starts from the very C-terminal residues for the EF-G G-domain. This reviewer thus recommends publication of the revised manuscript for publication in Nature Communications.

Tae-Young Yoon

Reviewer #4 (Remarks to the Author):

The manuscript by Xiuqi Chen and co-authors has been revised. Additional information has been added, which was missing in the original version of the manuscript. In sum, I feel that the authors have responded to the majority of the previous concerns either through additional experiments or changes to the text.

Response to reviewers' comments

In the response to the reviewers' comments on the revised version of our manuscript "Synthesis runs counter to directional folding of a nascent protein domain", reviewer comments are shown in black, our response is shown in blue font color.

Reviewer #1:

The authors have carefully revised the manu[script] and addressed my concerns.

Reviewer #2:

The authors addressed the reviewers' comments in a thorough manner with additional experimental data, which is remarkable given the current pandemic situation. In particular, the new observations that the construct 316RNC does not show a stable folding intermediate even under a prolonged observation time of ~300 s (Figure 6C and Suppl. Figure 4) shore up the authors' conclusion that the folding process starts from the very C-terminal residues for the EF-G G-domain. This reviewer thus recommends publication of the revised manuscript for publication in Nature Communications.

Reviewer #4:

The manuscript by Xiuqi Chen and co-authors has been revised. Additional information has been added, which was missing in the original version of the manuscript. In sum, I feel that the authors have responded to the majority of the previous concerns either through additional experiments or changes to the text.

We are delighted that the reviewers are satisfied with the changes and additions that were made to the manuscript during revision.

We wish to thank the reviewers again for their thorough evaluation and constructive comments on the original manuscript, which were instrumental in strengthening data presentation, interpretation and conclusions from our work.